# Backward Location and Travel Time Probabilities for Pollutants Moving in Three-Dimensional Aquifers: Governing Equations and Scale Effect

**Chaloemporn Ponprasit [1], Yong Zhang [1],\*and Wei Wei [2,3]**

1   Department of Geological Sciences, University of Alabama, Tuscaloosa, AL 35487, USA; cponprasit@crimson.ua.edu
2   School of Environment, Nanjing Normal University, Nanjing 210023, China; wwei@njnu.edu.cn
3   Jiangsu Center for Collaborative Innovation in Geographical Information Resource Development and Application, Nanjing 210023, China
\*   Correspondence: yzhang264@ua.edu

**Abstract:** Backward probabilities have been used for decades to track hydrologic targets such as pollutants in water, but the convenient deviation and scale effect of backward probabilities remain unknown. This study derived backward probabilities for groundwater pollutants and evaluated their scale effect in heterogeneous aquifers. Three particle-moving methods, including the backward-in-time discrete random-walk (DRW), the backward-in-time continuous time random-walk (CTRW), and the particle mass balance, were proposed to derive the governing equation of backward location and travel time probabilities of contaminants. The resultant governing equations verified Kolmogorov's backward equation and extended it to transient flow fields and aquifers with spatially varying porosity values. An improved backward-in-time random walk particle tracking technique was then applied to approximate the backward probabilities. Next, the scale effect of backward probabilities of contamination was analyzed quantitatively. Numerical results showed that the backward probabilities were sensitive to the vertical location and length of screened intervals in a three-dimensional heterogeneous alluvial aquifer, whereas the variation in borehole diameters did not influence the backward probabilities. The scale effect of backward probabilities was due to different flow paths reaching individual intervals under strong influences of subsurface hydrodynamics and heterogeneity distributions, even when the well screen was as short as ~2 m and surrounded by highly permeable sediments. Further analysis indicated that if the scale effect was ignored, significant errors may appear in applications of backward probabilities of groundwater contamination. This study, therefore, provides convenient methods to build backward probability models and sheds light on applications relying on backward probabilities with a scale effect.

**Keywords:** backward model; governing equation; scale effect; backward location probability; backward travel time probability

## 1. Introduction

Kolmogorov derived his well-known transport equations describing the probability density of random-walk particles in both jump and diffusion processes in 1931 [1,2]:

$$\frac{\partial G}{\partial t} = -\sum_{i=1}^{3} \frac{\partial}{\partial Y_i^*}(A_i^* \, G) + \frac{1}{2} \sum_{i,j=1}^{3} \frac{\partial^2}{\partial Y_i^* \, \partial Y_j^*} \left\{ \left[ B^* (B^*)^T \right]_{ij} G \right\} \tag{1}$$

$$\frac{\partial P}{\partial s} = -\sum_{i=1}^{3} A_i \frac{\partial P}{\partial X_i^*} - \frac{1}{2} \sum_{i,j=1}^{3} \left[ BB^T \right]_{ij} \frac{\partial^2 P}{\partial X_i \, \partial X_j} \tag{2}$$

where $G$ is the forward conditional probability density, i.e., the probability to find the particle in location $Y$ [L] at time $t$ [T], given it was in $X^*$ at time $s$ ($s < t$); $P$ is the backward

conditional probability density, i.e., the probability that the particle was in location $X$ at time $s$, given it is in $Y$ at time $t$; and the superscript $T$ represents the transpose of the matrix. Here $A$ and $A^*$ are the "drift" vectors $[LT^{-1}]$, and $B$ and $B^*$ are tensors $[LT^{-1/2}]$ defining the strength of diffusion along the backward and forward directions, respectively. Equation (1) is called the Kolmogorov forward equation or the Fokker–Planck equation, and Equation (2) is the Kolmogorov backward equation. During the past four decades, hydrogeologists have used the forward equation of Equation (1) (with a slightly different diffusive flux to account for the mass balance, see Section 3.1) to simulate the random walk of particles to calculate aqueous concentrations of solutes driven by advection and dispersion [3–9]. The corresponding numerical method is called the random-walk particle tracking (RWPT) method. The RWPT method is computationally appealing because it does not need any space discretization, does not suffer from numerical dispersion in problems dominated by advection, conserves the global mass balance automatically, and can be incorporated into any flow problem [5,9]. The continuum-based classical solvers, such as the finite element method, the finite difference method, and the method of characteristic models, do not have these advantages. Therefore, the high efficiency of the RWPT method promotes applications of the forward equation (Equation (1)) and its variant, which is adopted by this study.

Kolmogorov's backward equation (Equation (2)) has also been applied, although by relatively fewer users, to reverse problems in hydrology. For example, since Uffink [10] first applied Kolmogorov's backward equation to calculate the history of groundwater contamination, backward probability has been used to delineate well-head protection zones [11–13], calculate groundwater ages based on the concentrations of environmental tracers [14], evaluate the aquifer vulnerability [15,16], identify the groundwater pollutant source [17–20], and study the important influence of pumping on natural aquifer recharge [21]. These studies demonstrate that the application of backward probability has a high efficiency in backward problems for the case of non-point source contaminants and does not need initial conditions of solutes on model boundaries (as shown again by this study). These advantages are not present in the forward transport methods.

Further development is required in both the theoretical derivation and field application of backward probabilities of contaminants. There have been two main methods used to derive the governing equation of backward probabilities, namely, the adjoint method proposed by Kolmogorov [1,2] and expanded recently by Neupauer and Wilson [22–24] and Zhang et al. [25,26], and the traditional method of the Taylor series expansion of the Chapman–Kolmogorov equation initiated by Fokker and Planck [27]. However, four main questions remain for the backward probability models. *First*, methods that can conveniently reverse the forward transport models to their backward counterpart are still needed for hydrologists. *Second*, one outstanding question that limits further applications of backward probabilities is whether the vector $A$ and the tensor $B$ in Kolmogorov's backward equation must be divergence free. *Third*, current applications of backward probabilities are either limited to macroscopically homogeneous or simplified heterogeneous porous media, or one- or two-dimensional media [26]. The systematic behaviors of backward probabilities in three-dimensional natural geological media, such as regional-scale alluvial aquifer/aquitard systems, have not yet been reported in literature. *Fourth*, the above-mentioned limitations question the commonly used assumption that the scale effect of backward probabilities can be ignored. In general, for a short (1.5 m for a commonly used short-screen) and narrow well surrounded by highly permeable sediments, researchers assumed that local dispersion around the well screen is negligible compared with the regional-scale dispersion occurring between the well and the contaminant source (see for example the particle tracking transport modeling conducted by Weissmann et al. [14] using the middle interval of a well screen). Therefore, one conclusion is usually drawn and applied pervasively—the sample collected from either one interval, or the entire interval of a short screen, is representative of waters at any interval of the well screen. This assumption

neglects the scale effect of backward probabilities, and quantitative evaluations are needed to support or correct it.

This study tries to fill the four knowledge gaps mentioned above. We develop the theoretical basis, improve the calculation algorithm, extend the application areas of backward probabilities, and quantitatively evaluate the scale effect of backward probabilities of contaminants in heterogeneous aquifer systems (notably, all natural aquifers are heterogeneous) in three steps. First, three different methods of particle moving, including the backward-in-time discrete random-walk (DRW) method, the backward-in-time continuous time random-walk (CTRW) method, and the particle mass balance method, are proposed and applied to derive the governing equation of backward location probability (BLP) and backward travel time probability (BTTP) of contaminants in groundwater flow systems. We show for the first time that, by tracking backward in time, the widely used random walk and mass balance theories can conveniently lead to backward probability models. Next, an improved backward-in-time RWPT technique is applied to solve the backward probabilities numerically. Finally, the effects of diameter, length, and depth of the well screen on BLP and BTTP are explored for a three-dimensional heterogeneous medium and its equivalent homogeneous counterpart using numerical solutions.

## 2. Governing Equations of Backward Probabilities

Backward probabilities of a particle, including BTTP and BLP, provide the probability of a particle at certain previous time(s) and location(s) given that the probability of the particle at the current time and location is 1. Specifically, BTTP describes the time required for the particle to travel from a known location to an observation point or area, and BLP describes the possible positions of the particle at a known time or period in the past.

The motion of a particle is well known to be composed of two processes: one is driven deterministically by a drift vector, and the other is driven by a Gaussian noise (or a non-Gaussian noise shown by our previous work [25,26]). This motion can be described by the following nonlinear Langevin equation, which is a stochastic differential equation modeling random dynamics driven by deterministic and fluctuating forces [27]:

$$dX = A \, dt + B \, dW \tag{3}$$

where $W$ represents a Wiener process [$T^{1/2}$], and $dX$ denotes the particle's displacement during time $dt$. Interpreting Equation (3) in its integral version and applying the characteristics of Ito integration, one can use the following equation to describe the previous information of a particle given its current location and time:

$$X(t - \Delta t) - X(t) = -A \, \Delta t - B \, \Delta W \tag{4}$$

A reliable backward model is the prerequisite for reliable applications of BLP/BTTP. In the following sub-sections, we derive Kolmogorov's backward equation by solving the probability density function (PDF) of particles with displacements obeying Equation (4). For cross verification and extension purposes, three different methods are used.

### 2.1. Backward-in-Time Discrete Random-Walk (DRW) Method

The DRW method is first shown here because of its simplicity. For description simplicity, here we use a one-dimensional random walk (whose three-dimensional extension is straightforward). Let $P_n(l')$ denote the probability of a walker being at $l'$ after $n$ steps in a one-dimensional random walk. Assuming that the individual steps of the random walk are independent and identically distributed, we have the following backward-in-time recurrence relationship:

$$P_n(l') = \sum_{l=-\infty}^{+\infty} p(l - l') \, P_{n+1}(l) \tag{5}$$

where $p(l - l')$ denotes the probability that a walker who is currently at location $l'$, moved from location $l$. Let $\Delta$ be the step size and $e$ be the time interval between successive steps, and their limits toward zero; then, Equation (5) can be re-written as:

$$P(x, s) = \sum_{l=-\infty}^{+\infty} p(l - l') \, P[x + (l - l')\Delta, \ s + e] \tag{6}$$

where $P(x, s)$ is the PDF for the walker located at $x$ at time $s$.

Assuming $P[x + (l - l')\Delta, \ s + e]$ is differentiable once with respect to $s$ and twice with respect to $x$, we can re-write Equation (6) using a formal Taylor expansion:

$$P(x, s) = \sum_{l=-\infty}^{+\infty} p(l - l') \left\{ P(x, s) + \Delta(l - l') \frac{\partial P(x, s)}{\partial x} + e \frac{\partial P(x, s)}{\partial s} + \frac{\Delta^2}{2}(l - l')^2 \frac{\partial^2 P(x, s)}{\partial x^2} + O\left(e + \Delta e + \Delta^2\right) \right\} \tag{7}$$

Because $\sum_{l=-\infty}^{+\infty} p(l - l') = 1$, Equation (7) can be re-arranged to:

$$-e \frac{\partial P(x, s)}{\partial s} = \Delta \frac{\partial P(x, s)}{\partial x} \sum_{l=-\infty}^{+\infty} (l - l') p(l - l') + \frac{\Delta^2}{2} \frac{\partial^2 P(x, s)}{\partial x^2} \sum_{l=-\infty}^{+\infty} (l - l')^2 p(l - l') + O\left(e + \Delta e + \Delta^2\right) \tag{8}$$

When $e \to 0$ and $\Delta \to 0$, the truncation error $O\left(e + \Delta e + \Delta^2\right) \to 0$ and becomes negligible.

We already know that:

$$\lim_{\Delta, \, e \to 0} \frac{\Delta}{e} \sum_{l=-\infty}^{+\infty} (l - l') p(l - l') = a \tag{9}$$

$$\lim_{\Delta, \, e \to 0} \frac{\Delta^2}{2e} \sum_{l=-\infty}^{+\infty} (l - l')^2 p(l - l') = \frac{1}{2} b^2 \tag{10}$$

where $a$ is the $x$ component of $A$, and $b$ is the $x$-diagonal component of $B$ [5,27]. They are also the two parameters used in the nonlinear Langevin equation [27–29]. Dividing Equation (8) by $e$ and inserting Equations (9) and (10) in Equation (8) leads to the following backward probability model:

$$-\frac{\partial P(x, s)}{\partial s} = a \frac{\partial P(x, s)}{\partial x} + \frac{b^2}{2} \frac{\partial^2 P(x, s)}{\partial x^2} \tag{11}$$

*2.2. Backward-in-Time Continuous Time Random-Walk (CTRW) Method*

The CTRW method is considered here because it is theoretically stricter than the DRW method. There have been hundreds of papers discussing Brownian motion using CTRW (see for example, the review in [30]), but most of them focus on the forward-in-time movement of particles. An in-depth introduction of the CTRW method can be found in the classical work of Metzler and Klafter [31]. The Galilei variant/invariant assumptions, which might be disputable, sometimes are used to directly add an advection term to the diffusion equation [31]. Here we extend the CTRW method in Metzler and Klafter [31] for backward-in-time solutions and eliminate the Galilei variant/invariant assumptions.

The backward PDF is:

$$P(x, \tau) = \int_0^\tau \eta(x, \tau') \Psi(\tau - \tau') \, d\tau' \tag{12}$$

where $\tau$ represents the backward increase in time with $\tau = \Delta s > 0$, $\eta(x, \tau')$ denotes the PDF for particle reaching $x$ at time $\tau'$, and $\Psi(\tau - \tau')$ is the PDF for particles without movement during the period of $\tau'$ to $\tau$. We use $x_d$ and $t_d$ to represent the detection location and time for forward-in-time formulations. The backward time is $s = t_d - \tau$, where $0 < s < t_d$. The

corresponding Fourier transform (with the symbol of hat ˆ) and Laplace transform (with the symbol of bar −) of Equation (12) is:

$$\overline{\hat{P}}(k,u) = \overline{\hat{\eta}}(k,u)\,\overline{\Psi}(u) \tag{13}$$

We then derive the expression of $\overline{\hat{\eta}}(k,u)$ and $\overline{\Psi}(u)$. Considering the decoupled jump length and waiting time PDF [31]:

$$\eta(x,\tau) = \int_{-\infty}^{\infty}\int_{0}^{\tau}\eta(x',\tau')\,p(\tau - \tau')\,\lambda(x - x')\,dx'\,d\tau' + \delta(x - x_d)\delta(\tau - 0) \tag{14}$$

where $\delta$ is the Dirac delta function. Equation (14) is equal to:

$$\begin{aligned}
\eta(x,\tau) &= \eta(x',\tau') * p(\tau) * \lambda(x) + \delta(x - x_d)\delta(\tau) \\
&= \eta(x',\tau') * p(\tau) * \lambda(x) + P_0(x - x_d)\delta(\tau)
\end{aligned} \tag{15}$$

where $p$ is the waiting time PDF, $\lambda$ is the jump length PDF, and the symbol "*" denotes convolution. Taking both the Fourier and Laplace transforms for Equation (15), and then solving for $\overline{\hat{\eta}}(k,u)$, we have:

$$\overline{\hat{\eta}}(k,u) = \frac{\hat{P}_0(k)}{1 - \overline{p}(u)\hat{\lambda}(k)} \tag{16}$$

Because $\Psi(\tau) = 1 - \int_{0}^{\tau}p(\tau')d\tau'$, the corresponding Laplace transform of $\Psi$ is:

$$\overline{\Psi}(u) = \frac{1}{u} - \frac{1}{u}\overline{p}(u) \tag{17}$$

Inserting Equations (16) and (17) into Equation (13) leads to:

$$\overline{\hat{P}}(k,u) = \frac{\hat{P}_0(k)}{1 - \overline{p}(u)\hat{\lambda}(k)}\frac{1 - \overline{p}(u)}{u} \tag{18}$$

To solve Equation (18), we first assume a Poissonian waiting time PDF [31]:

$$p(\tau) = \frac{1}{T}\,\exp\left(-\frac{\tau}{T}\right) \tag{19}$$

where $T$ is the mean waiting time and $T > 0$. We then assume that the particle jump size satisfies a Gaussian PDF (and the particle has different probabilities of directional particle movement for multi-dimensional extension). This allows particles to have a non-zero mean jump length $x_0$:

$$\lambda(x) = \frac{1}{\sqrt{4\pi\,\sigma^2}}\,\exp\left[-\frac{(x - x_0)^2}{4\,\sigma^2}\right] \tag{20}$$

where $\sigma$ is the standard deviation of the random jump size.

The Laplace/Fourier transform of $p(\tau)$ and $\lambda(x)$ is $\overline{p}(u) = 1/(Tu + 1)$ and $\hat{\lambda}(k) = \exp\left(-k^2\sigma^2 + ix_0k\right)$, respectively. By inserting the first-order Taylor expansion of $\overline{p}(u)$ and the second-order expansion of $\hat{\lambda}(k)$ into Equation (18), we have:

$$\overline{\hat{P}}(k,u) = \frac{T\hat{P}_0(k)}{Tu + \left(\frac{x_0^2}{2} + \sigma^2\right)k^2 - \left(\frac{x_0^2}{2} + \sigma^2\right)Tuk^2 - ix_0k + ix_0kTu} \tag{21}$$

Assuming a long time and large space limit as in Metzler and Klafter [31], we know that $\left(\frac{x_0^2}{2} + \sigma^2\right)Tuk^2$ is a higher order term than $\left(\frac{x_0^2}{2} + \sigma^2\right)k^2$, and $ix_0kTu$ is a higher order

term than $ix_0k$. Then we obtain the following equation using the reverse Laplace and Fourier transforms:

$$\frac{\partial P(x,\tau)}{\partial \tau} = \frac{b^2}{2}\frac{\partial^2 P(x,\tau)}{\partial x^2} + a\frac{\partial P(x,\tau)}{\partial x} \tag{22}$$

According to the relationship between the backward time and the increase in backward time, we can transfer Equation (22) to Equation (11), proving the backward probability model of Equation (11).

The above-mentioned random-walk-based methods indicate that the governing equation of the backward-in-time PDF has the same form as Kolmogorov's backward equation. This result is not limited to one dimension because random walk methods can be extended to three dimensions [2,32,33].

### 2.3. Method of Conservation of Particle Mass

The mass-balance method is proposed here for three-dimensional expansion. It is a reasonable assumption that each particle moves randomly in multiple dimensional spaces. For simplicity, we consider particle transport in the $X$ direction first, and then we combine the spatial movements of each particle. Let $M_i$ be the number of particles in cell $i$; then, the particle density in this cell is given by:

$$P_i = M_i / U_i \tag{23}$$

where $U_i$ is the volume of cell $i$ [$L^3$]. Assuming that the particle generally moves from cell $i$ to cell $i+1$ under ambient conditions, then the particle number flux from cell $i+1$ to cell $i$ per unit area and per unit time in the backward-in-time process is [34]:

$$F_x = \frac{\left(\frac{1}{2}+\phi_i\right)M_i - \left(\frac{1}{2}-\phi_{i+1}\right)M_{i+1}}{\omega} = \left[\left(\frac{1}{2}+\phi_i\right)P_i - \left(\frac{1}{2}-\phi_{i+1}\right)P_{i+1}\right]R\,\Delta x \tag{24}$$

where the parameter $\phi_i$ represents the difference in probabilities when particles jump forward and backward along $X$-axis, so $\phi_i > 0$; $\omega$ is the area of cell normal to $X$-axis [$L^2$]; $R$ is the number of jumps per unit time for each particle [$T^{-1}$]; and $\Delta x$ is the cell length [$L$]. Using the following Taylor series approximation:

$$P(x+\Delta x,\,s) = P(x,\,s) + \frac{\partial\,P(x,s)}{\partial x}\,\Delta x + O\left(\Delta x^2\right) \tag{25}$$

we can then rewrite Equation (24) as:

$$F_x = (\phi_i + \phi_{i+1})R\,\Delta x\,P_i - \left(\frac{1}{2}-\phi_{i+1}\right)R\,\Delta x^2\frac{\partial P_i}{\partial x} - \left(\frac{1}{2}-\phi_{i+1}\right)O\left(\Delta x^3\right)R \tag{26}$$

When $\Delta x \to 0$, it is obvious that $(\phi_i + \phi_{i+1})\,R\,\Delta x \to a$. According to Fick's law [30], $-\left(\frac{1}{2}-\phi_{i+1}\right)R\,\Delta x^2 \to b^2/2$. Equation (26) hence becomes:

$$F_x = aP + \frac{b^2}{2}\frac{\partial P(x,\tau)}{\partial x} \tag{27}$$

For conservative solutes, the total number of particles remains stable during jumping events. The conservation of particle mass also means the conservation of the number of particles (which carry the solute mass or backward probability). Substituting Equation (27) into $-\partial P/\partial s = \nabla\cdot F$, the mass conservation equation, and then expanding it to three dimensions, one obtains:

$$\frac{\partial P}{\partial s} = -\nabla\cdot\left(AP + \frac{1}{2}BB^T\,\nabla P\right) \tag{28}$$

This formula extends Kolmogorov's backward equation by showing that the vector *A* and tensor *B*, which control the advective and diffusive displacement of particles, do not necessarily have to be divergence free. This conclusion is consistent with the results of Neupauer and Wilson [24] using the rigid, but mathematically complex, sensitivity-based adjoint approach. In the following we discuss whether this extension is reasonable in applications.

### 3. The Backward-in-Time RWPT Technique to Solve the Backward Probabilities

Parameters in the particle transport equation are determined by the analogy between particle jumps (i.e., Kolmogorov's equation) and solute transport equations. This analogy is the key to solving backward probabilities using the backward equation [10].

#### 3.1. Parameter Identification

At first, the particle probability density in the forward equation can be transformed to the solute aqueous concentration in contaminant transport equation. The analogy between forward equation and the ADE, therefore, can determine the parameters controlling the forward movements of particles. The well-known ADE for non-reactive solute is:

$$\frac{\partial(nC)}{\partial s} = -\nabla \cdot (n\boldsymbol{V}C) + \nabla \cdot (n\boldsymbol{D}\,\nabla C) \tag{29}$$

where $C$ is the aqueous concentration $[ML^{-3}]$, $\boldsymbol{V}$ is the average groundwater velocity vector $[LT^{-1}]$, and $\boldsymbol{D}$ is the local hydrodynamic dispersion tensor $[L^2T^{-1}]$. Parameters between the forward equation of Equation (1) and the ADE (Equation (29)) are related by:

$$G = nC/m \tag{30}$$

$$\boldsymbol{A}^* = \boldsymbol{V} + \nabla \boldsymbol{D} + \boldsymbol{D}\,\nabla n/n \tag{31}$$

$$\boldsymbol{B}^*(\boldsymbol{B}^*)^T = 2\boldsymbol{D} \tag{32}$$

where $m$ represents the mass of particles $[M]$, and $\nabla \boldsymbol{D}$ denotes the gradient of $\boldsymbol{D}$.

Then the solution of the backward equation can be transformed to the solution of the forward equation [10]. When the backward equation takes the form of Equation (2) or Equation (11), the main parameters for the backward equation (Equation (11)), the forward equation (Equation (1)), and the ADE (Equation (29)) are related by:

$$\boldsymbol{A} = -\boldsymbol{A}^* + \boldsymbol{B}\boldsymbol{B}^T = -\boldsymbol{V} + \nabla \boldsymbol{D} \tag{33}$$

$$\boldsymbol{B}\boldsymbol{B}^T = \boldsymbol{B}^*(\boldsymbol{B}^*)^T = 2\boldsymbol{D} \tag{34}$$

$$s = t_d - t \tag{35}$$

The prerequisites of the above-mentioned formulas are (a) the flow is steady-state, and (b) the porosity of the porous media is constant. Herein, when the vector *A* and tensor *B* controlling the movement of particles are divergence free, the corresponding groundwater flow is steady-state, and the porosity is spatially and temporally invariant.

When the backward equation takes the form of Equation (28), we derive the relationships of main parameters among the backward equation (Equation (28)), the forward equation (Equation (1)), and the ADE (Equation (29)):

$$\boldsymbol{A} = -\boldsymbol{A}^* + \boldsymbol{B}\boldsymbol{B}^T = -\boldsymbol{V} + \nabla \boldsymbol{D} - \left(\frac{1}{n}\right)\nabla(n\boldsymbol{D}) \tag{36}$$

$$\boldsymbol{B}\boldsymbol{B}^T = \boldsymbol{B}^*(\boldsymbol{B}^*)^T = 2\boldsymbol{D} \tag{37}$$

$$s = t_d - t \tag{38}$$

which are similar to Equations (33)–(35), except for the first line. This formula is valid for any groundwater flow condition and is not restricted to constant porosity (notably, when flow is steady state and the medium porosity is constant, Equation (36) reduces to Equation (33), as expected). Therefore, when the gradients of vector *A* and tensor *B* are non-zero, corresponding flow and porosity values are more representative of natural conditions. In addition, it is noteworthy that the tensor *D* takes the same form in all three equations (backward equation, forward equation, and ADE), whereas the vector *A* does not have this property. It is consistent with Arnold's conclusion that only the diffusion operator is self-adjoint [32]. Notably, if the diffusive jumps follow a non-Gaussian distribution (such as super-diffusion along preferential flow paths or fractured rock mass), the diffusion operator is no longer self-adjoint because preferential jumps of particles are now direction dependent and non-symmetric (different from the symmetric, direction-independent Fickian diffusion considered in this study); for details, please see our recent work for backward models for anomalous diffusion [25,26].

### 3.2. Numerical Techniques and Verifications

First, the initial and boundary conditions of the original forward transport model must be modified to fit the backward conditions. The main modifications include: (a) the reversion of particle source/sink terms; (b) the no-flux boundary ($VC - D \nabla C = 0$, also named as the 3rd-type boundary) in forward transport problems transfers to a no-gradient boundary ($D \nabla P = 0$, also named as the 2nd-type boundary, representing a free exit boundary) in backward probability problems, and vice versa (the same as in ref. [23]); and (c) if the no-flux boundary of forward solute transport is also the no-flux boundary of groundwater flow model, we use the same no-flux boundary for backward probabilities. In RWPT solutions [5,6], a particle-absorbing boundary is set to represent the no-gradient boundary, and a particle-reflecting boundary to represent the no-flux boundary for backward probabilities.

Then, we refine the releasing manner of particles at the initial time to account for potential differences in fluxes entering a well screen at different intervals. The initial mass of each particle released around the screen is proportional to the corresponding flux at the same direction and location. This algorithm accurately models transport around a three-dimensional well in a heterogeneous porous medium and is more reasonable than the uniform-releasing method applied, for example, by ref. [12].

The approach is illustrated using a one-dimensional, semi-infinite domain that mimics the flow of a production well, and these numerical results are compared to the analytical solutions provided by Neupauer and Wilson [23], Equations (10) and (11). The domain was extended from $0 \leq x < \infty$, with the well at $x = 0$ and an instantaneous point source of contaminant at $x \geq 0$. The actual flow was from right to left (well) at a seepage velocity of 0.24 ft/day, porosity 0.25, and dispersivity 10 ft. As demonstrated by Figure 1, the RWPT method can reliably simulate backward probabilities. This test indicates that (a) at a single point in an aquifer, the BLP or BTTP is not a single value but a wide distribution (a skewed normal distribution in homogeneous cases); and (b) the simulated BTTP contains more noise. It is well known that the solutions of particle tracking methods become smoother if the time step is smaller and the particle number is larger.

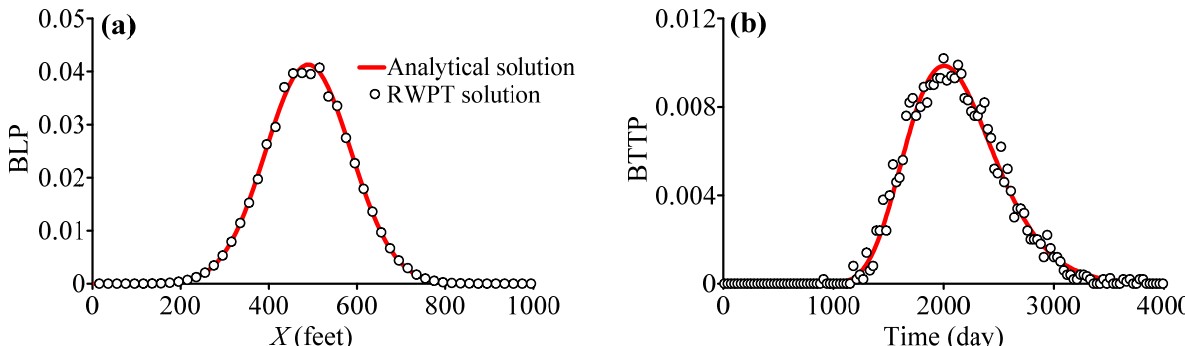

**Figure 1.** Analytical versus numerical solutions of backward location probability (BLP) and backward travel time probability (BTTP). (**a**) Plot of BLP from a pumping well, showing the probability at *s* = 2000 days for all possible locations of contaminant sources. (**b**) Plot of BTTP from a pumping well, for an upstream contaminant source at *x* = 500 feet. The time step of numerical simulations is 1 day.

## 4. Quantitative Evaluation of Scale Effects of BLP and BTTP

Here, we use three main steps to numerically evaluate the scale effects of backward probabilities. The first step is to build geologic models to represent regional-scale subsurface heterogeneity using transition probability/Markov chain-based geostatistical methods [35]. Next, the groundwater velocities in the simulated aquifers are calculated using the finite difference code MODFLOW from the U. S. Geological Survey [36]. Finally, backward location and travel time probabilities are calculated using the backward-in-time RWPT methods proposed in Section 3.

### 4.1. Hydrologic Condition, and Flow and Transport Parameters of the Study Area

One heterogeneous model was constructed for the purpose of evaluating the scale effect of backward probabilities in three-dimensional, complicated heterogeneous media (Figure 2a). The Markov chain model for the study site was built by Carle [37] and Fogg et al. [38], and then developed by LaBolle et al. [39]. It represents an alluvial depositional environment dominated by fine sediments, which is located beneath the Lawrence Livermore National Laboratory, California. Four hydrofacies, namely, debris flow, flood plain, levee, and channel (whose properties are listed in Table 1), were recognized by thousands of meters of cores and drillers' logs.

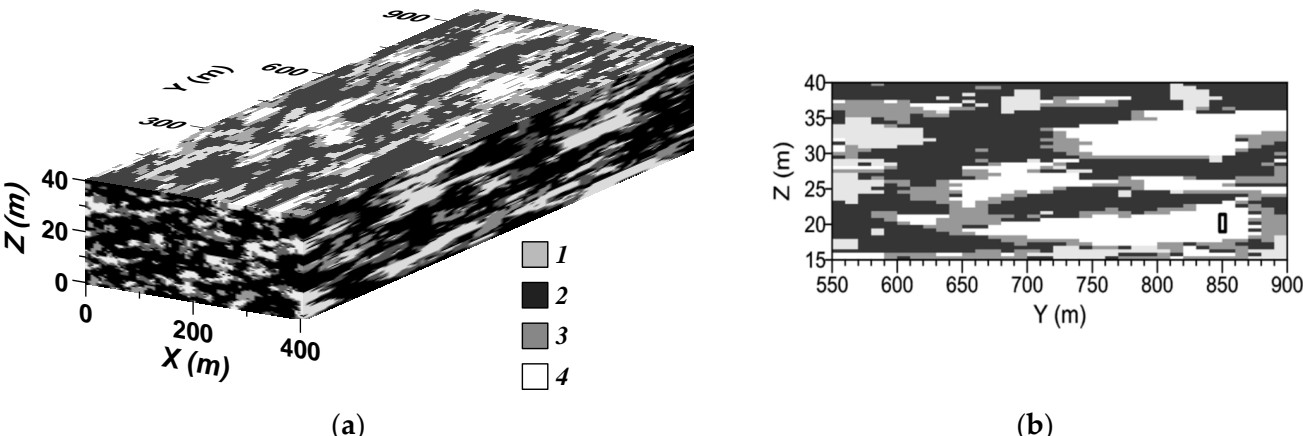

(**a**)                                                                                     (**b**)

**Figure 2.** The heterogeneous model (**a**) and one cross-section (along *Y* direction) located at the middle of the model domain (**b**). The legends 1, 2, 3, and 4 in (**a**) represent debris flow, floodplain, levee, and channel hydrofacies, respectively (which are the same as "ID" listed in Table 1). The small rectangle in the lower right of (**b**) denotes the well screen. The size of this screen is exaggerated to make its location clear.

**Table 1.** Hydrofacies' properties for the study site. In the legend, $L_{Strike}$, $L_{Dip}$, and $L_{Vertical}$ represent the mean length of each facies along the strike, dip, and vertical directions, respectively; "Proportion" denotes the volumetric global proportion of hydrofacies (from Carle [37]); "ID" represents the identity number of the hydrofacies in the geostatistical model; and $K$ denotes the mean hydraulic conductivity calibrated by Fogg et al. [38].

| Hydrofacies | ID | $L_{Strike}$ [m] | $L_{Dip}$ [m] | $L_{Vertical}$ [m] | Proportion [%] | $K$ [m/d] |
|---|---|---|---|---|---|---|
| Debris flow | 1 | 8 | 24 | 1.1 | 7 | $4.32 \times 10^{-1}$ |
| Floodplain | 2 | 27 | 67 | 2.1 | 26 | $4.32 \times 10^{-6}$ |
| Levee | 3 | 6 | 20 | 0.8 | 19 | $1.73 \times 10^{-1}$ |
| Channel | 4 | 10 | 50 | 1.3 | 48 | $5.18 \times 10^{0}$ |

Although its global proportion is only 18%, channel facies, the most permeable material, forms the main aquifers by interconnecting spatially [38]. To make the heterogeneity model appropriate for this study, we made some modifications. First, a 2.5 m thick, sandy sediment was preserved as hard conditional data. It is located near the downgradient boundary and represents a well surrounded by highly permeable materials (see Figure 2b). Then, the model domain in LaBolle et al. [39] was extended upstream, to include the full transport path of solutes from the water table to the well. The cell size of the geologic model is 5, 10, and 0.5 m in the depositional strike, depositional dip, and vertical directions, respectively (Figure 2).

The size of cells at and around the monitoring well in the groundwater flow model (and the following solute transport model) was refined to be 0.1, 0.1, and 0.5 m in the depositional strike, dip, and vertical directions, respectively. The boundary conditions of the steady-state flow model are similar to those used by LaBolle et al. [39]. The top boundary represents a constant recharge with a rate of 0.034 m/year (representing the average net annual recharge to groundwater; see [39]), and others are general head boundaries. The hydraulic conductivity, $K$, for each facies (listed in Table 1) has already been calibrated by Fogg et al. [38] by modeling field pumping tests, so it was used here unchanged. The top of the simulation roughly corresponds to the observed water table [39].

In the backward particle-tracking model, we set the top and upgradient boundaries to be particle-absorbing boundaries, and others to be particle-reflecting boundaries. Because the dispersion term in the backward equation explains the uncertainty of probabilities as we go back further in time and location, it has the same purpose as the dispersion term in the forward ADE, also pointed out by ref. [12]. Hence, in our backward particle-tracking model, the dispersivity (0.01 m), molecular diffusion coefficient ($5.2 \times 10^{-5}$ m$^2$/d), and effective porosity (0.35) are the same as those used by LaBolle et al. [39] in their forward transport model.

A homogeneous and anisotropic model, which is equivalent to the heterogeneous model (Figure 2a), was built for comparison purposes. Anisotropic values for $K$ of 0.108, 0.449, and $3.61 \times 10^{-3}$ m/d for the depositional strike, depositional dip, and vertical directions, respectively, are the up-scaling values of the heterogeneous model according to Darcy's law. Effective $K$ values in all three directions were determined through separate simulations of flow in the depositional strike, dip, and vertical directions [13]. Similar boundary conditions to those described above were used for this homogeneous simulation.

*4.2. Results of the Calculated Backward Probabilities*

In the homogeneous medium, the calculated BLP of contaminants in groundwater collected at different depths along the 2.5 m long screen of the monitoring well have similar main characteristics (Figure 3), except for the following subtle differences. The distribution area of BLP at the water table moves more slowly upstream when the sampling location gets deeper. The corresponding area of distributions increases slightly at the same time. The shapes of BLP distributions at the water table during whole transport periods are

concentric ellipses, regardless of well length and the location of screened intervals within that well. Additionally, the area of the high probability zone, which is in the middle of the ellipses, decreases when the sampling interval moves to the bottom of the well. On the contrary, the area of low probability zone increases at the same time, implying the higher uncertainty (and/or a larger upstream area) with a longer backward time.

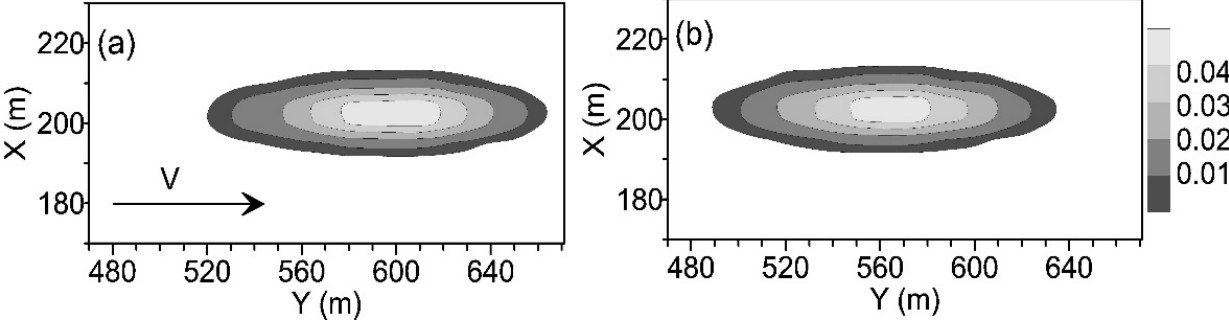

**Figure 3.** The simulated BLP at the water table for the homogeneous and anisotropic model (described at the end of Section 4.1). The effective time is the entire backward transport time when all particles hit the water table. The sampling location is the top interval ($z$ = 21.4 m) (**a**) and the bottom interval ($z$ = 19.1 m) (**b**) of the well screen shown in Figure 2b. The arrow in (**a**) represents the regional-scale flow direction of groundwater. The $z$ value used in all figures is consistent with the $z$-coordinate showing in Figure 2.

The corresponding BTTP and mean travel time in this homogeneous aquifer change regularly with the variation in sampling depths (Figure 4a). When the sampling location becomes deeper, the BTTP distribution shifts to older zones along the time axis (Figure 4a) (because of the larger travel distance) with only slight fluctuations in the distribution width and peak. The corresponding mean travel time follows an almost linear trend as the depth of the sample interval increases (shown by the hollow circles in Figure 5b). The average acceleration rate of mean travel times along this 2.5 m long well screen is 3.36 year/m, resulting in an 8-year difference for ground water sampled at the top interval (with the depth of $z$ = 21.5 m) and the bottom interval ($z$ = 19.0 m) of this well.

Calculations show that the horizontal size of the screen has minimum influence on backward probabilities. The first two moments of BTTP in the homogeneous model are almost identical when the diameter of the well screen increases from 0 to 0.2, 0.4, and 0.6 m (where the mean backward travel time is shown by the triangles in Figure 5a). The similar behavior was also found in the heterogeneous media (shown by the circles and diamonds in Figure 5a).

In the heterogeneous model, the BTTP is, however, sensitive to the variation in sampling depths inside of the screen (Figure 4b). When sampling at the top, middle, and bottom intervals of a same screen, the behaviors of corresponding BTTP at these different intervals are quite different. The top interval receives recharge within a relatively short period (about 15 years), whereas the bottom interval receives its recharge in a 50-year-long range without apparent peaks. The simulated mean backward travel time also fluctuates obviously with the variation in sampling depths (Figure 5b). The mean backward travel time at the bottom interval is 30 years older than that at the top interval. This discrepancy is even larger than the mean travel time at the top interval.

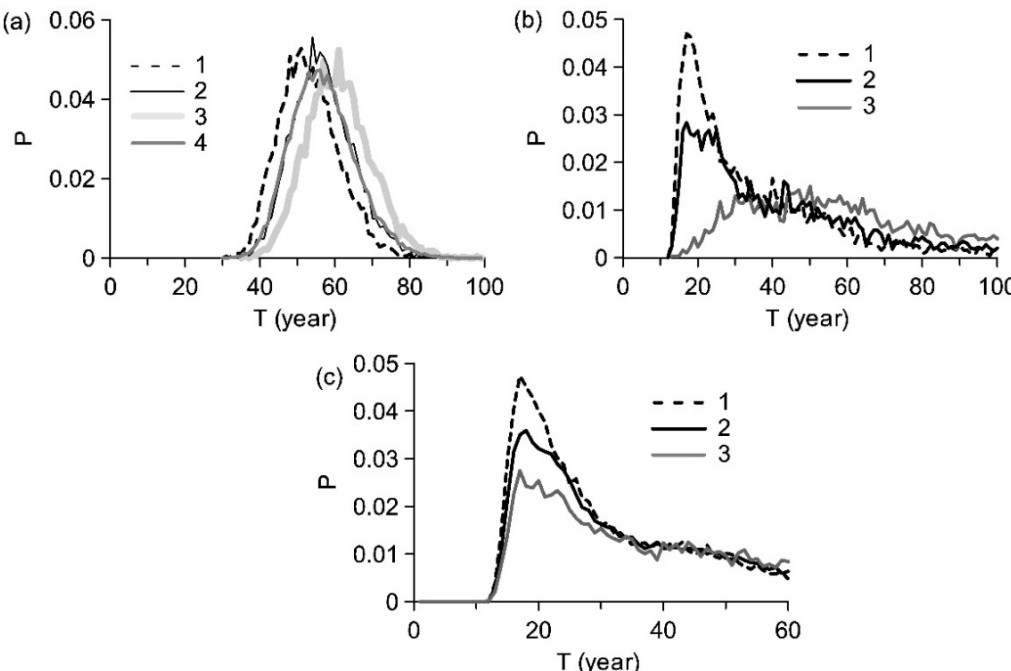

**Figure 4.** The simulated backward travel time probability (BTTP) for groundwater collected from the top interval (line 1, $z$ = 21.4 m), middle interval (line 2, $z$ = 20.25 m), bottom interval (line 3, $z$ = 19.1 m), and the entire 2.5 m long interval (line 4) for an equivalent homogeneous model (**a**) and its original heterogeneous counterpart (**b**). In (**c**), the sampling location is the whole screen, and the length of this screen is 0.5 m (line 1, $z$ = 21.0–21.5 m), 1.5 m (line 2, $z$ = 20.0–21.5 m), and 2.5 m (line 3, $z$ = 19.0–21.5 m), respectively.

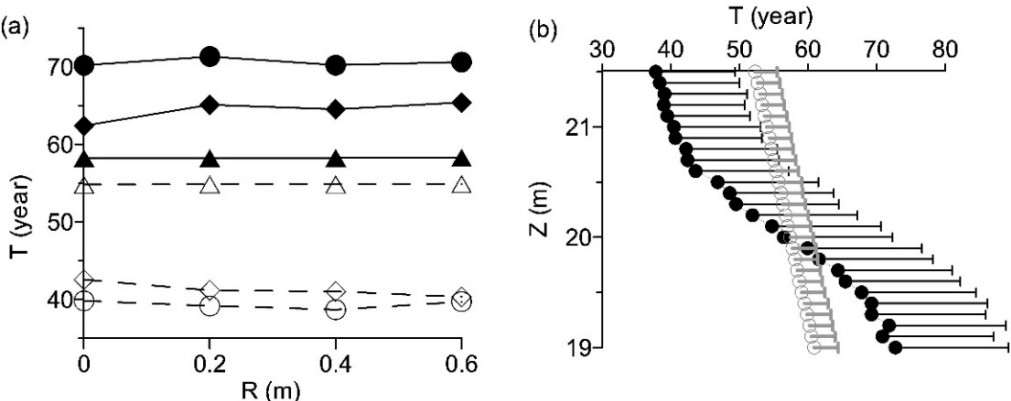

**Figure 5.** (**a**) The simulated first moment of backward travel time versus the diameter of the well screen. The sampling location is the bottom intervals ($z$ = 19.0–19.5 m, shown by the solid triangle) and the top intervals ($z$ = 21.0–21.5 m, shown by the hollow triangle) of an effective homogenous model; and the bottom intervals ($z$ = 19.5–20.0 m, the solid diamond) and the top intervals ($z$ = 20.5–21.0 m, the hollow diamond) of the original heterogeneous model; and the bottom intervals ($z$ = 19.0–19.5 m, the solid circle) and the top intervals ($z$ = 21.0–21.5 m, the hollow circle) of the original heterogeneous model. (**b**) The simulated first moment (i.e., the mean) of backward travel time at each point along the whole well screen located inside the heterogeneous model and its effective homogeneous model. The horizontal bar represents the standard deviation of age distributions along the positive direction. The solid and hollow circles represent the corresponding results of the heterogeneous model and its effective homogeneous counterpart, respectively.

In the heterogeneous model, the vertical length of the well screen also plays an important role in BTTP. When the screen length increases from 0.5 to 1.5 and 2.5 m, the

resultant peak of BTTP distribution decreases by ~50% (Figure 4c). This is because the proportion of old components for water packages collected through the whole screen increases with the increase in screen lengths.

The characteristics of BLP may explain the behaviors of BTTP mentioned above (Figure 6). The water packages entering the bottom interval of the screen originate from further upstream zones and further horizontal zones than the water packages entering at the top interval. In other words, the water packages captured by the bottom interval generally have longer travel distances and larger recharge areas. This discrepancy results not only in a longer tail of backward travel times, but also in a lower peak in probability distribution with a larger width for probability mass conservation. Therefore, in a complicated heterogeneous aquifer such as the typical alluvial setting, if the vertical lengths of the screens of monitoring wells are different, or if the vertical sampling depths in this well screen are different, both the BLP and the BTTP may be quite different, even for a short screen surrounded by highly permeable sediments.

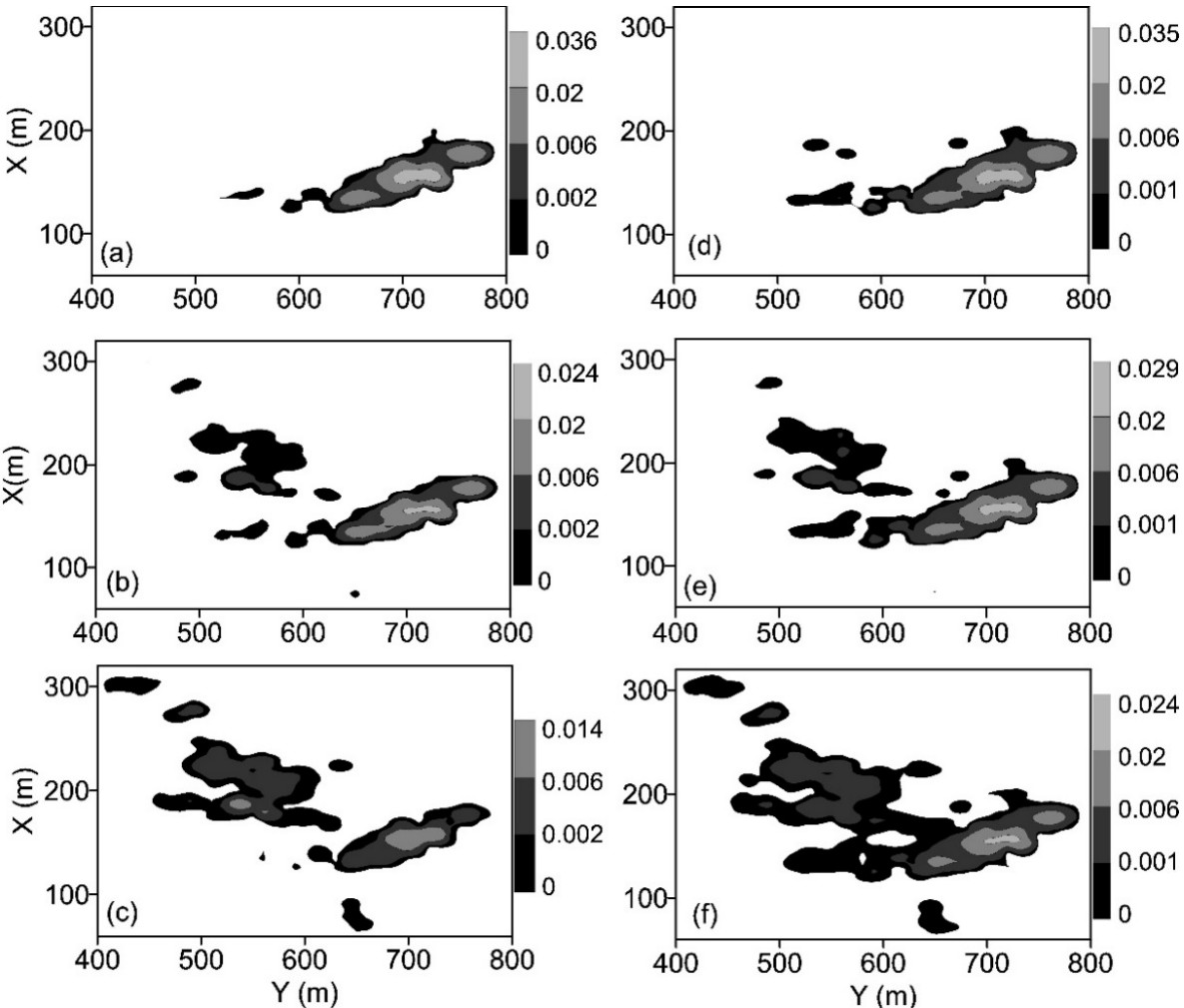

**Figure 6.** The simulated backward location probability of groundwater collected from wells (the small rectangle in Figure 2b) located inside the heterogeneous model shown in Figure 2a. The effective sampling time is the complete transport time for all particles until they hit the water table. The sampling location is located at the top interval ($z = 21.4$ m) (**a**), the middle interval ($z = 20.3$ m) (**b**), the bottom interval ($z = 19.1$ m) (**c**), respectively. The samples are also collected through the whole screened intervals with a length of 0.5 m ($z = 21.0–21.5$ m) (**d**), 1.5m ($z = 20.0–21.5$ m) (**e**), and 2.5m ($z = 19.0–21.5$ m) (**f**), respectively. To show better the BLP peak position, a slightly different grey scale (for the maximum level) is used in these plots.

## 5. Discussion

The major finding of this work (i.e., BLP and BTTP are sensitive to the vertical interval and length of well screens) has profound meanings for real-world applications where BTP and BTTP are used as critical indices. Specifically, the scale effect of backward probabilities can result in strong spatial and temporal variations in measured concentrations of groundwater samples, which thus raises serious questions in the current applications of backward probabilities, including the monitoring and evaluation of groundwater quality, identification of groundwater pollutant sources, assessment of aquifer vulnerability, and delineation of well-head protection zones. For instance, as indicated by Figure 7, the normalized concentrations of contaminants measured at different depths along a short screen (2 m long) may vary up to 1 order of magnitude due to the scale effect of backward probabilities. This strong variation results in the following dilemma. If the monitoring network of groundwater quality is too sparse (a common scenario), it is difficult to capture variations in concentration within such a small local scale, and then it is most likely to miss the main characteristics of contaminant plumes. On the contrary, if the well screen is relatively long, a small amount of water collected from specific interval(s) of the well may not represent the aquifer where the well is. In both cases, it is important to evaluate the measurements based on the exact sampling location to obviate any misleading data. The method proposed by this study can be developed to assist field works, such as the design of monitoring networks.

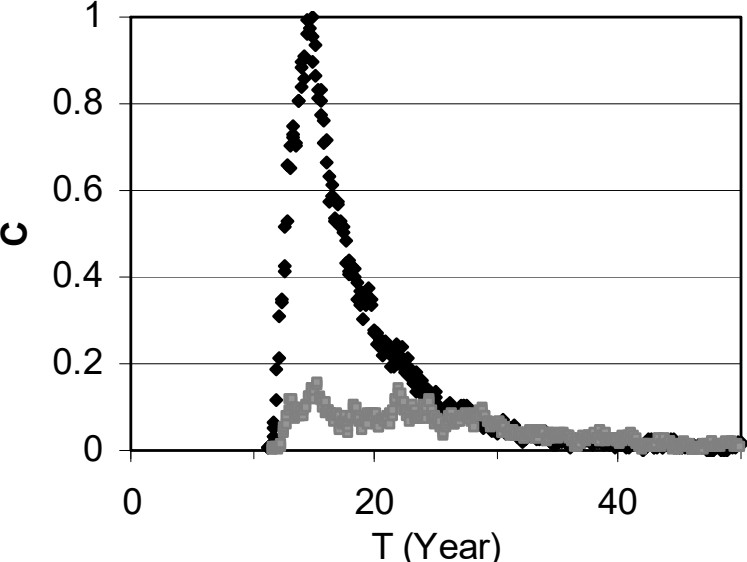

**Figure 7.** The simulated normalized concentration at the top and bottom intervals of the well screen (0.5 m long for each interval) with an instantaneous source ($x$ = 175~185 m, $y$ = 711 m, and $z$ = 39.01 m) near the water table. The dark and the light points represent the concentrations at the top and bottom intervals, respectively.

As mentioned above, researchers simplify the heterogeneous aquifers for many reasons, such as lack of data or scale limitations. Comparisons in this study indicate the heterogeneous model has significantly different backward probabilities compared to the equivalent homogeneous and anisotropic models. However, our hydrogeologic interpretive skills have been strongly influenced by homogeneous conceptual models. We need to better understand the limitations of replacing a real-world heterogeneous medium by a homogeneous model when investigating backward probabilities, because the homogeneous model may be misleading.

This study therefore answered the four backward probability-related questions raised in Introduction. *First*, three particle-moving methods, namely, the backward DRW, backward CTRW, and particle mass balance, conveniently converted the forward-in-time trans-

port model to its backward counterpart. *Second*, the backward conversion showed that the vector ***A*** and the tensor ***B*** in Kolmogorov's backward equation need not to be divergence free. *Third*, the backward PDF properties were systematically analyzed for pollutants moving in a three-dimensional alluvial aquifer, whose nuance cannot be fully captured by its "equivalent" homogeneous model. *Fourth* and most importantly, extensive numerical experiments revealed the strong (vertical) scale effect of backward probability, challenging the commonly used assumption that the scale effect of backward probability is negligible for regional-scale natural aquifers.

Future extensions of this work are needed. For example, subsurface hydrodynamics and heterogeneity distributions influence the scale effect of backward probabilities and thus require further investigation. *First*, the main hydrodynamic conditions affecting the variations in backward probabilities include boundary conditions and transport parameters used in the simulation. Boundary conditions include the rate of recharge applied to the top boundary and horizontal flux from the upgradient boundary. Our preliminary results (not shown here) revealed that a larger recharge rate from the top boundary and/or a smaller flux from the upgradient boundary will cause smaller variations in backward probabilities along the well. In the models used by this study, the most important transport parameter is the molecular diffusion coefficient. A larger molecular diffusion coefficient may enhance leakage recharge from sediments having low permeability, resulting in more old components in the backward travel time probability distribution. *Second*, variation in the heterogeneity structure, such as the correlation length and the proportion of hydrofacies, may result in different preferential paths for both water and solute. Therefore, the heterogeneity structure may play an important role in the scale effect of backward probabilities. This topic will be further investigated in a future paper. *Finally*, the most important factor in a real-world application that may change the simulation results of this study is the actual distributions of depositional materials around the screened well. Sediments having low permeability may form mixed layers, such as clay laminae, within the highly permeable materials around the screen. The existence of low-permeability materials can enhance the difference in water intakes at different depths of the screen, and then enhance the scale effect of backward probabilities. One possible means to address this issue is to build and analyze multiple different but equally possible realizations for each scenario of hydrofacies models using the geostatistical tool applied above. The uncertainty of the calculated backward probabilities caused by the above factors deserves further research.

## 6. Conclusions

This study tried to fill the knowledge gaps of backward probabilities by building the governing equations and evaluating the scale effect of backward location and travel time probabilities for pollutants moving in a three-dimensional aquifer. Three main conclusions were drawn.

First, the governing equation of backward location probability and backward travel time probability cross-verified Kolmogorov's backward equation and extended the theoretical basis of backward probabilities. The improved backward RWPT technique extended the application of backward probabilities to more complex, three-dimensional, heterogeneous alluvial settings. The groundwater flow field is not limited to steady-state conditions and the media do not have to have constant porosity values (see Equation (36), for example, where the velocity can be time dependent and the porosity can change in space).

Second, numerical experiments indicated that the backward probabilities are not sensitive to the well screen diameter, because the horizontal scale of the aquifer is much larger than the diameter of a well screen (hundreds of meters verses ~$10^{-1}$ to $10^0$ m in this study). Therefore, a well can be simplified to be a vertical line inside an aquifer system during numerical modeling. Numerical simulations conducted by LaBolle et al. [39] supported this conclusion by showing that the main behaviors of plume migrations were not significantly influenced by a limited variation in the initial horizontal location of contaminants.

Third, the backward location and travel time probabilities of groundwater contaminants can be significantly influenced by the variations in the vertical lengths of the well screen or the depths of sampling points along the well screen in complex heterogeneous aquifers. The results of this study showed that the backward probabilities of contaminants from one depth inside a 2.5 m long screen surrounded by highly permeable materials cannot represent the backward probabilities of contaminants in water packages entering the screen through another depth. Although the local dispersion around the well screen is negligible compared with the regional-scale dispersion occurring between the well and the source, groundwater can reach individual intervals of the same screen from different pathways connecting the water table and the well. Thus, the backward probabilities can change vertically, resulting in a scale effect of backward probabilities. The scale effect of backward probabilities may strongly affect real-world applications relying on BLP/BTTP.

**Author Contributions:** Conceptualization, Y.Z.; Methodology, C.P. and Y.Z.; Software, Y.Z.; Validation, C.P., Y.Z. and W.W.; Formal Analysis, C.P. and Y.Z.; Investigation, Y.Z.; Resources, Y.Z.; Data Curation, Y.Z. and C.P.; Writing-Original Draft Preparation, C.P. and Y.Z.; Writing Review & Editing, C.P. and Y.Z.; Visualization, Y.Z.; Supervision, Y.Z.; Project Administration, Y.Z.; Funding Acquisition, W.W. and Y.Z. All authors have read and agreed to the published version of the manuscript.

**Funding:** W.W. was partially supported by the National Natural Science Foundation of China (No. 41931292). Y.Z. was partially supported by the Alabama Center of Excellence (Project of Groundwater 2070 in Baldwin County, Alabama under a Changing Climate and Threatened by Seawater Intrusion: From Sustainability to Vulnerability). This paper does not necessarily reflect the views of the funding agencies.

**Data Availability Statement:** The date presented in this study are available on request from the corresponding author.

**Conflicts of Interest:** The authors declare no conflict of interest. The funders had no role in the design of the study; in the collection, analyses, or interpretation of data; in the writing of the manuscript, or in the decision to publish the results.

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
