# Peer review of "Backward Location and Travel Time Probabilities for Pollutants Moving in Three-Dimensional Aquifers: Governing Equations and Scale Effect"

_water, doi:10.3390/w14040624_

Round 1

Reviewer 1 Report

This manuscript describes the development of a new travel time distribution model for pollutants in 3D heterogeneous aquifers and investigates the effects of limited monitoring well extensions on backward travel times. This is a welcome research objective, even of first models of this king were introduced already in 1931 by Kolmogorov. The manuscript gives an accurate account of developments since that early work and formulates 4 research question at the end of the introduction that still are unanswered despite almost 90 years of work. Then, an extended model approach if given, and a 3D heterogeneous model domain is generated to test it. The results show that quite large differences are found in backward travel time distributions when the supposed screen length of a monitoring well are varied. This is interesting and shows the small simplifications in previous works could have rather large impacts on the results.

The paper is well written, novel and of good quality and clarity. It can be published almost in its present form. The reviewer would like nevertheless to ask for some revisions:

  • In the discussion section, the authors should resume the four unanswered research questions from the introduction and state, one by one, how the present work has found answers to each question.
  • Figure 1: the designations a and b are missing, and it looks like the order of appearance of the two graphs is wrong (with respect to the legend).
  • Table 1 and Figure two; the facies are not in the same order. It is suggested to add the numbers 1-4 used in the figure in Table 1 in an additional column, then the legend of figure 2 can make reference to table 1. Order Table 1 according to 1-4 in Figure.
  • Figure 3: Why compare anisotropic to homogeneous? Why not present 4 cases: homogeneous vs heterogeneous, and isotropic vs anisotropic?
  • Figure 3: If point 4 is not adopted: in legend, state that anisotropic is figure 3a, and homogeneous is Figure 3b.
  • 6: Why not present the shades of grey all on the same scale?

Author Response

1) In the discussion section, the authors should resume the four unanswered research questions from the introduction and state, one by one, how the present work has found answers to each question.

Reply: We thank the reviewer for the very helpful comments that improved the presentation of this work. We added the answer to each of the four questions in Discussion (lines 475~485).

2) Figure 1: the designations a and b are missing, and it looks like the order of appearance of the two graphs is wrong (with respect to the legend).

Reply: We corrected Fig. 1 in the revised manuscript.

3) Table 1 and Figure two; the facies are not in the same order. It is suggested to add the numbers 1-4 used in the figure in Table 1 in an additional column, then the legend of figure 2 can make reference to table 1. Order Table 1 according to 1-4 in Figure.

Reply: Done. We added this column in Table 1 (by re-ordering the 4 facies) and then referred it in Figure 2.

4) Figure 3: Why compare anisotropic to homogeneous? Why not present 4 cases: homogeneous vs heterogeneous, and isotropic vs anisotropic?

Reply: We explained this anisotropic and homogeneous model in the last paragraph in section 4.1 (lines 348-355). This model is anisotropic because the effective hydraulic conductivity K is direction dependent (which is common for a 3-d heterogeneous aquifer with a longer correlation of lithologies along the depositional strike/dip direction than the vertical direction), and homogeneous because the effective K remains constant in space (which is needed for upscaling). We emphasized this point in the revised manuscript (please see the revised Caption of Figure 3). We did not present these 4 cases in Fig. 3 because (1) the heterogeneous cases were already shown in the following figures, and (2) the effective homogeneous model was not isotropic due to the direction-dependent K (as explained in section 4.1: “Anisotropic values for K of 0.108 m/d, 0.449 m/d, and 3.6110-3 m/d for the depositional strike, depositional dip, and vertical directions, respectively”).

5) Figure 3: If point 4 is not adopted: in legend, state that anisotropic is figure 3a, and homogeneous is Figure 3b.

Reply: Fig. 3a and Fig. 3b used the same anisotropic and homogeneous model introduced in the last paragraph in section 4.1; so, they are both homogeneous and anisotropic.

6) 6: Why not present the shades of grey all on the same scale?

Reply: The reason that we used a slightly different grey scale was to improve the visualization for the BLP peak. We added this explanation in the revised manuscript (lines 460~461). Figs. 6a~6c used the same grey scale (0.002 and 0.006) except for the maximum level, because (1) each plot had a different peak value for the backword location PDF, and (2) we wanted to emphasize the position of the peak BLP since it represents the most likely source position. Due to the same reason, Figs. 6d~6f used the same grey scale (0.001, 0.006, and 0.02) except for the maximum level.

Reviewer 2 Report

Attached

Author Response

1) In the abstract lines 15-19. Use abreviations for; the back-ward-in-time discrete random-walk (DRW), the backward-in-time continuous time random-walk (CTRW) and all abbreviations mentioned in the abstract as well.

Reply: We thank the reviewer for the very helpful comments that improved the presentation of this work. In the revised manuscript (lines 16~17), we used these abbreviations.

2) Edit the two paragraphs in Line 116-119 and also line 252-253.

Reply: Done. We edited these two paragraphs.

3) In line 119, add more elaboration about the Langevin equation (27).

Reply: Done. We added the explanation for the Langevin equation in the revised manuscript (lines 118-120).

4) In Line 235-241 and (Eqs 33-35) the prerequisites for these equations are Steady-State flow conditions and porous media; In Lines 500-501 in the conclusion section; you said “The groundwater flow field is not limited to steady-state conditions and the media do not have to have constant porosity values”.

Reply: To obviate any confusion, we added the explanation for the transient flow and spatially variable porosity in Conclusion (lines 521~522). The “steady-state flow condition” and “constant porosity” in Line 235-241 were defined only for Equations (30)~(32), while the transient flow and spatially variable porosity were considered by Equations (36)~(38).

5) Line 328 more elaboration and citation for why you assigned the constant recharge by 0.034.

Reply: We added the explanation and citation for this recharge (lines 335-336).